

# Escaping the benthos with Coral Reef Arks: effects on coral translocation and fish biomass

Jessica Carilli[1],[*], Jason Baer[2],[*], Jenna Marie Aquino[2], Mark Little[3], Bart Chadwick[4], Forest Rohwer[2], Gunther Rosen[1], Anneke van der Geer[2], Andrés Sánchez-Quinto[2], Ashton Ballard[2] and Aaron C. Hartmann[3]

[1] Naval Information Warfare Center Pacific, San Diego, CA, United States
[2] San Diego State University, San Diego, CA, United States
[3] Harvard University, Cambridge, MA, United States
[4] Coastal Monitoring Associates, San Diego, CA, United States
[*] These authors contributed equally to this work.

Corresponding author
Jessica Carilli,
jessica.c.carilli.civ@us.navy.mil

## ABSTRACT

Anthropogenic stressors like overfishing, land based runoff, and increasing temperatures cause the degradation of coral reefs, leading to the loss of corals and other calcifiers, increases in competitive fleshy algae, and increases in microbial pathogen abundance and hypoxia. To test the hypothesis that corals would be healthier by moving them off the benthos, a common garden experiment was conducted in which corals were translocated to midwater geodesic spheres (hereafter called Coral Reef Arks or Arks). Coral fragments translocated to the Arks survived significantly longer than equivalent coral fragments translocated to Control sites (*i.e.*, benthos at the same depth). Over time, average living coral surface area and volume were higher on the Arks than the Control sites. The abundance and biomass of fish were also generally higher on the Arks compared to the Control sites, with more piscivorous fish on the Arks. The addition of Autonomous Reef Monitoring Structures (ARMS), which served as habitat for sessile and motile reef-associated organisms, also generally significantly increased fish associated with the Arks. Overall, the Arks increased translocated coral survivorship and growth, and exhibited knock-on effects such as higher fish abundance.

## INTRODUCTION

Coral reef ecosystems are declining globally due to local and global stressors including overfishing, pollution, and climate change (*Eddy et al., 2021*). Most reef mitigation and restoration efforts have focused on protecting and rebuilding coral communities, due to the role of corals as ecosystem engineers. Such projects often rely on some form of coral translocation; for example, corals are moved off of piers to natural reef sites to mitigate damage (*Dickenson, McNeilly & Marx, 2002*). Corals are often fragmented and grown in nurseries, then outplanted to natural or artificial reef sites for restoration (*Bayraktarov et al., 2020*). These projects have varying success (*Boström-Einarsson et al., 2020*;

*Hein et al., 2020*), in part because transplanting corals to sites with poor environmental conditions is likely to fail unless the source of the poor conditions is addressed (*Ferse, Hein & Rölfer, 2021*).

Given that many environmental stressors causing coral reef decline are large-scale and unlikely to be remediated in the near future (*e.g.*, ocean warming), the Coral Reef Arks approach was designed to provide an interim solution to enhance the survival of corals, study the successional patterns of reef communities, and determine whether Arks may help surrounding areas recover ecosystem functions (*Baer et al., 2023*). The midwater Arks create suitable habitat in a location with better abiotic conditions, including higher light availability, flow speeds, and dissolved oxygen, and lower microbial biomass and abundance (*Baer et al., 2023*) than the ocean bottom (hereafter referred to as the benthos, to include the non-living ocean floor and associated biota) (*Webb et al., 2021*), and provide corals translocated to this habitat with reef-associated biota to support ecosystem services necessary to promote coral and reef survival. These services include grazing to reduce competition with algae, nutrient remineralization, water filtering, and defense against corallivores (*Stella et al., 2011*; *Nelson, Wegley Kelly & Haas, 2023*). Reef-associated species are translocated to the Arks using Autonomous Reef Monitoring Structures (ARMS), which provide habitat and passively collect a significant fraction of reef diversity from natural reef sites (*e.g.*, *Ransome et al., 2017*; *Rohwer & Hartmann, 2020*) before being transferred to the Arks.

During the nursery stage for coral gardening projects, corals are often elevated off the benthos with tables or ropes and nets suspended by buoys, providing corals with improved water quality and resulting in higher survival and growth rates compared to benthic nurseries (*e.g.*, *Shafir, Van Rijn & Rinkevich, 2006*; *Nedimyer, Gaines & Roach, 2011*). These nurseries are intended as a temporary holding site for corals prior to affixing them to the benthos, often require significant maintenance, and do not create a complex reef system to support coral growth in the long term, which is the ultimate goal of restoration. In contrast to growing corals in isolation for short periods, Arks are intended to provide the same or more beneficial water quality conditions as nurseries, while creating an artificial reef for corals to permanently reside. To do this, the Arks are placed shallow enough to meet the light requirements of corals and other photosynthetic organisms, off the benthos, and far enough from shore to reduce exposure to runoff and other local impacts. Furthermore, Arks are seeded with cryptic biodiversity via ARMS to support coral health and replace human maintenance (*e.g.*, algae and corallivore removal) with nature-based solutions (*e.g.*, herbivores and predators). As such, Arks are designed to meet the Coral Restoration Consortium priorities to "Support a holistic approach to coral reef ecosystem restoration" and to "Increase restoration efficiency," by outplanting a range of coral species and genotypes as well as non-coral species (*Vardi et al., 2021*). Depending on site conditions and requirements and logistical support, Arks could theoretically be maintained in the midwater indefinitely or relocated to the seafloor on a suitable anchoring structure after an initial midwater period; however these longer term outcomes are yet to be tested.

Here, we describe two Arks structures deployed in Vieques, Puerto Rico. Stony corals were translocated to the Arks in two stages 6 months apart, first without, and then with an accompanying transfer of seeded ARMS units. Corals were also translocated to two benthic Control sites akin to traditional coral outplanting approaches during each stage. Biotic and abiotic metrics were subsequently tracked at multiple monitoring timepoints. This article presents results from the first five monitoring timepoints, spanning approximately 19 months on the Arks and Control sites to address three related hypotheses: 1) corals translocated to the Arks will survive longer and have greater skeletal and/or tissue growth than corals translocated to the benthic Control sites, 2) turf and macroalgae cover around corals on the Arks will be lower than at the benthic Control sites, and 3) fish abundance and biomass associated with the Arks will be greater than fish associated with the benthic Control sites.

We present macroorganismal data to test these hypotheses here. We previously showed that the Arks and Control sites differ in abiotic and microbial conditions, and thus differ in their theoretical suitability for coral survival, as intended in our experimental design (*Baer et al., 2023*). The Vieques Arks have higher water flow rates, higher light levels likely due at least in part to reduced sedimentation, lower diel variation in dissolved oxygen, lower concentrations of dissolved organic carbon (DOC), more viruses relative to their microbial prey, and smaller microbial cell sizes compared to the Control sites (*Baer et al., 2023*). The Arks conditions are similar to those found on coral-dominated reefs throughout the world, while those on the Control sites are more closely aligned with low coral cover, degraded, and "microbialized" reefs (*Haas et al., 2016*; *Silveira et al., 2023*).

## METHODS

### Experimental design

#### Site design: Coral Reef Arks

Arks are midwater, positively buoyant, 2.4 m (8 ft) diameter geodesic spheres tethered to the seafloor. Regulatory approvals to conduct this demonstration were obtained in conjunction with the Vieques Restoration Project, particularly the National Marine Fisheries Service Programmatic Biological Opinion (OPR-2017-00026). In November 2021, two Arks were deployed offshore approximately 2 miles to the west of Vieques Island, Puerto Rico (Figs. 1A and 1B), within part of the Navy's unexploded ordnance (UXO) remediation site 16 (UXO16). The seafloor in this area is 16.7 m (55 feet) deep and consists of sand with patches of rubble and macroalgae such as *Padina spp*. and *Halimeda spp*. in the immediate area. A mapping survey of Vieques underwater habitat classified the Arks deployment area as sand, with coral reef and hardbottom/pavement habitat located approximately 100 m south of the Arks site (*Bauer & Kendall, 2010*). Arks were installed using a set of three helical sand anchors and a multipoint bridle system described in *Baer et al. (2023)*, following specific guidelines for work within a UXO site. Once installed, the top of Ark1 and Ark2 was located at approximately 7.6 m (25 ft) and 7.3 m (24 ft) below the water surface, respectively. The two Arks were separated by approximately 50 m.
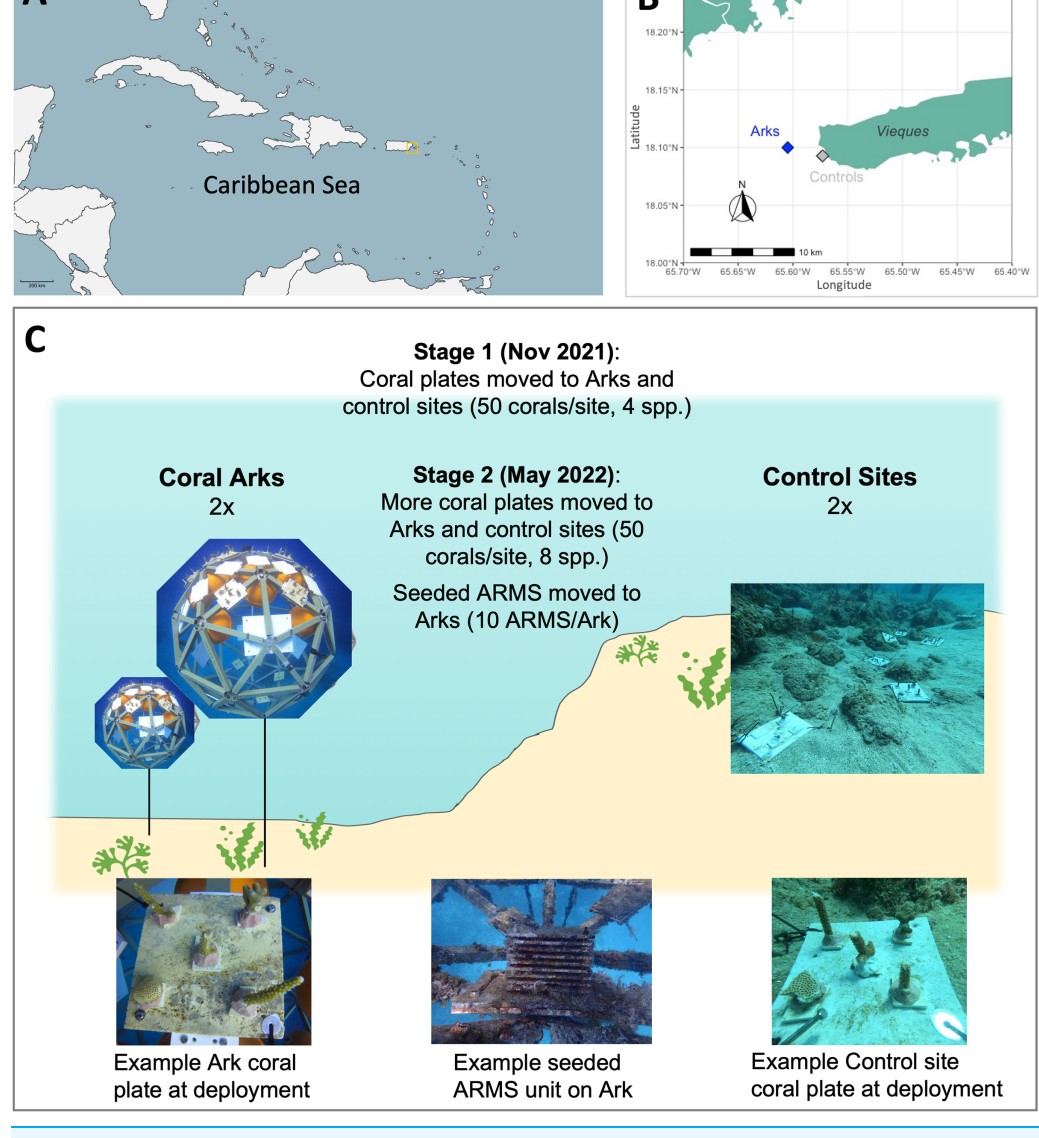

**Figure 1 Maps of (A) regional setting and (B) treatment sites for Arks and Control sites, and (C) schematic representation of experimental design.**

Additional details regarding building and deploying Arks can be found in *Baer et al. (2023)*.

## Site design: Control sites

Two Control sites were established at similar depths as the tops of the Arks (7.6 and 6.4 m/ 25 and 21 ft water depth, respectively), to compare the Arks approach to the traditional approach of translocating corals to the benthos at appropriate depths, which only occur relatively close to shore (compared to the offshore locations of the Coral Arks). While this experimental design did not allow the separation of distance-from-bottom from distance-to-shore factors, the overall experiment intended to holistically compare the Arks approach, which allows placement of corals in optimized conditions away from coastal

runoff, to traditional coral outplanting, which is constrained by available hardbottom at appropriate depths occuring along coastlines. The two Control sites were also located off the west coast of Vieques Island within another portion of UXO16 (Fig. 1B). The two Control sites were separated by approximately 25 m. The habitat in this area was classified as reef hardbottom characterized by colonized pavement, linear reef, and aggregated patch reef habitats (*Bauer & Kendall, 2010*). Qualitatively, the sites are dominated primarily by rock colonized by turf and macroalgae, stony corals (mainly in the genera *Orbicella*, *Siderastrea*, *Porites*, and *Diploria*), gorgonians, fire corals, and other sessile invertebrates, with scattered patches of sand and seagrass found at the deeper fringes of the sites.

### Site design: ARMS seeding

ARMS are three-dimensional structures made of PVC plates and stainless-steel hardware that create a standardized area of substrate to passively collect reef communities *via* natural recruitment and growth (www.oceanARMS.org). Thirty ARMS were placed on the benthos in the vicinity of the Control sites off the west coast of Vieques, located between about 8 to 14 ft depth and close to living coral assemblages. ARMS were secured to the benthos in sets of five using rebar stakes and cable ties to link the ARMS and concrete bags as weighted anchors (*Baer et al., 2023*). ARMS were left to accumulate coral reef cryptic biodiversity for a 1-year "seeding" period before they were moved to the Arks. No ARMS were moved to the control sites, as these sites were established adjacent to natural reef communities already replete with the biota the ARMS accumulated.

## Coral sourcing and translocation

Corals of opportunity were used for this experiment, with approval from Puerto Rico's Department of Natural and Environmental Resources (DNER), permit number O-*VS*-PVS15-SJ-01233-20092021. Corals were translocated to the Arks and Control sites in two cohorts 6 months apart (November 2021 and May 2022; Table S1). Approximately half of the corals were sourced from a NOAA coral nursery called Palominos, off the east coast of the main island of Puerto Rico during both time periods (*Acropora cervicornis*, *Porites porites* for both cohorts and *Orbicella spp.* in May 2022), and from metal debris (a barge and pipes) in Bahía de Jobos, Puerto Rico, slated for removal by DNER in November 2021 (*Porites porites* and *Siderastrea radians*). Additional corals of opportunity were obtained from rubble fields and a spalling concrete boat ramp on the south side of Mosquito Pier, Vieques, in May 2022 (*Porites furcata*, *Porites astreoides*, *Siderastrea siderea*, and *Agaricia sp.*). After collection, all corals were held in plastic bins with seawater (refreshed intermittently) or placed in plastic milkcrates suspended underwater beneath a small boat dock at Mosquito Pier. Corals were then fragmented and attached to numbered, unfinished limestone tiles (termed "coral plates") with a mixture of epoxy (*Aquastik Coralline Red*, *Two Little Fishies*) and superglue (*Seachem*). This attachment method was selected based on literature review and lab-based trials of different attachment methods.

Coral fragments were distributed such that individual nubbins of the same species or fragments from the same parent colony were placed on different coral plates and would be deployed to both the Arks and Control sites, providing an even balance of coral species and

genets between the two treatments. The following data were recorded for each coral fragment on each coral plate: species, genet, source site, date collected, approximate depth collected, date attached to coral plate, height, maximum horizontal dimension, horizontal dimension 90 degrees to maximum, number of branches if applicable (including number of branches with intact apical tips for *Acropora cervicornis* corals), and general health of the fragment (healthy, pale, bleached). Fewer than five collected corals had lesions consistent with stony coral tissue loss disease (SCTLD). Though SCTLD infection was not confirmed, these corals were not used on coral plates out of an abundance of caution.

Coral plates were attached with cable ties at a temporary holding site established in a rubble field on the south side of Mosquito Pier comprised of upside down plastic milkcrates, weights, and cinderblocks until plates were deployed to either an Ark or Control site. Corals remained on coral plates in the temporary holding location off Mosquito pier for variable time periods ranging from 0–9 days. While at the temporary holding location, corals were visually checked daily, and any accumulated fine sediment on the plates was fanned off. Attachment panels for coral plates were built into the Arks design and structure. At the Control sites, locations for coral plates were selected by a certified scientific diver to cluster coral plates relatively closely, as on the Arks structures, while avoiding areas that would impact living corals, native seagrass beds, or critical habitat for corals, and avoiding deep sand that might smother or scour the corals on the tiles. Divers then installed 2–4 stainless steel anchor points (camping spikes or lag bolts) into the benthos to which the coral plates were later attached.

Coral plates were deployed to either an Ark or a Control site by transferring them to the deployment site in bins of seawater on the shaded deck of a dive boat, and to the deployment site in milk crates. Coral plates were secured to either one of the Arks or to the benthos at one of the Control sites using stainless steel hardware and/or cable ties (Fig. 1C). The site, date, angle of deployment from horizontal, and condition of corals on the plates were recorded for each coral plate deployed.

### ARMS translocation

The Arks were monitored for the 6 months following coral translocation (stage 1), without the presence of seeded ARMS. In May 2022, ARMS were transferred to Arks (10 to each Ark) to seed the Arks with reef biodiversity (stage 2). ARMS were covered in a fine mesh to retain motile organisms, removed from the benthos, and brought to the surface. Each ARMS was individually placed in seawater-filled bins on the boat and kept in the shade during transit from the ARMS seeding site to the Arks (*Baer et al., 2023*). At the Arks, each ARMS was hand-carried from the boat to the Arks on SCUBA and attached to a pre-installed attachment plate built into the Arks. The ARMS were secured to the Arks with stainless steel hardware and zip ties, then the mesh bag was removed (Fig. 1C).

### Monitoring
#### *Coral survival and growth*
Data were collected at the Arks and Control sites at preplanned monitoring timepoints, immediately following the installation of the Arks (time 0), then approximately every

**Table 1 Summary of fish surveys completed (method and (number) of surveys)[1].**

| Treat. | # | Nov 2021 | Feb 2022 | May 2022 | Aug 2022 | Dec 2022 | Jun 2023 |
|--------|---|----------|----------|----------|----------|----------|----------|
| Ark | 1 | GoPro (1) | GoPro (1) | *In situ* (1) | *In situ* (2) | *In situ* (1) | *In situ* (2) |
| | 2 | GoPro (1) | GoPro (1) | *In situ* (1) | *In situ* (2) | *In situ* (1) | *In situ* (2) |
| Control | 1 | – | – | – | *In situ* (1) | *In situ* (1) | *In situ* (2) |
| | 2 | GoPro (1) | GoPro (1) | GoPro (1) | *In situ* (2) | *In situ* (1) | *In situ* (1) |

**Note:**
[1] Treat. indicates experimental treatment.

3 months for the first year, then another 7 months to span a total of about 19 months. At each monitoring timepoint, the following data were recorded *in situ* for each coral fragment: height, maximum horizontal dimension, horizontal dimension 90 degrees to maximum, number of branches if applicable, and general health (percent of living tissue that appeared healthy, pale, bleached, or diseased). If applicable, the percent of the entire fragment that had suffered partial mortality was also recorded. This data collection approach follows guidance from the NOAA Coral Reef Restoration Monitoring Guide (*Goergen et al., 2020*), with the addition of three-dimensional measurements to allow estimates of both living coral volume and surface area.

### Fish abundance, biomass, and diversity

Fish associated with the Arks and Control sites were observed and recorded from GoPro video footage and/or direct observations in the field (Table 1). In both cases, observations were based on approximately 10–15 min of video or direct observations at each site. All fish captured in a given video were identified to species, binned into various estimated size classes, and the number of fish in each estimated size class were counted. For *in situ* observations, stationary size estimates and counts were made to capture larger pelagic-associated fish, followed by closer-up mobile observations to record smaller and/or cryptic fish. The video approach proved more time intensive to accurately identify fish species, so this approach was replaced entirely with direct observations starting in August 2022. However, qualitatively, the methods produced comparable results, so the data collected at all timepoints are included here and considered representative of the site fish conditions at the monitoring timepoints. The focus of this effort was to capture the abundance and biomass of fish that were ecologically associated with either the Arks or the Control sites; therefore, although some large schools (100–300 individuals) of forage fish (such as sardines) were observed passing near the Arks, these were not enumerated. Similarly, nurse sharks that were observed around the Arks anchoring system were also not enumerated.

The trophic role of each species of fish observed was categorized based on literature references, in particular *Sandin & Williams (2010)*. Fish biomass was estimated using length-weight relationships published in Fishbase (*Froese & Pauly, 2023*), using the formula $W = a * L^b$, where $W$ is weight in grams, $L$ is length in cm (calculated as the midpoint of bins used for size estimates), and $a$ and $b$ are coefficients describing the relationship between length and weight for different fish species. Coefficients were mostly

obtained using the R package *rfishbase* or were manually retrieved from Fishbase if they were not included in the Fishbase length-weight table, but were estimated using Bayesian analysis of all length-weight measurements for fishes with similar body shapes (*Froese, Thorson & Reyes, 2014*).

### Turf and macroalgae on coral plates

At each monitoring timepoint, top-down photographs were collected of each coral plate. These images were used to visually estimate percent cover of turf algae and/or macroalgae for the portion of the coral plates not occupied by living corals. In cases where algae cover on the Control site plates accumulated sediment, this turf-consolidated sediment was also counted as turf/macroalgal cover. This metric was the strongest predictor of overall coral reef ecological function in a large-scale meta-analysis by *Silveira et al. (2023)*.

## Data analysis

Data analysis was conducted using R (Version 4.3.1) and RStudio statistical software (Version 2023.06.1 + 524; *R Core Team (2023)*). Because coral plates were deployed in two stages, time-since-deployment was used for coral analyses instead of calendar-time. To allow comparisons between stage 1 and stage 2 corals, time-since-deployment was approximated as 3 months (stage 1: November 2021 to February 2022, stage 2: May 2022 to August 2022), 6 months (stage 1: November 2021 to May 2022, stage 2: May 2022 to December 2022), 9 months (stage 1: November 2021 to August 2022), 12 months (stage 1: November 2021 to December 2022, stage 2: May 2022 to June 2023), and 19 months (stage 1: November 2021 to June 2023).

### Coral survival and growth

Coral survival was tracked and assessed using survival analysis methods to compare the length of time corals survived between treatments (Arks *vs*. Control sites). Here, loss of corals *via* death was considered the main event of interest and was scored categorically at each timepoint, with each coral nubbin assigned a 0 if at least part of the coral colony was alive (death had not occurred), or a 1 if the coral was completely dead. A separate categorical variable was used for missing corals that had broken off the plates between monitoring timepoints and for which the status (live or dead) at that timepoint was unknown. A coral could have been missing due to the epoxy failing or due to physical contact with the fragment which caused it to break off. Coral survival (in weeks since deployment) was visualized using a Kaplan-Meier survival plot, where missing corals and those that were still alive at the last monitoring timepoint are 'censored', indicating that the event (death) did not occur for the time period the subject was tracked, but it is unknown after that time whether or not the event occurred. In addition, a competing risks analysis was conducted, in which survival was coded as 0, and the events "death" and "missingness" were coded as 1 and 2, respectively, allowing assessment of the relative cumulative risk to coral survival based on the likelihood of dying or falling off coral plates. Differences in survival outcomes between treatments were statistically compared using log-rank tests and Gray's tests conducted in R software using the *survival* package.

**Table 2 Equations used to estimate living surface area and volume of corals.**

| Coral morphology | Volume formula | Surface area formula |
|---|---|---|
| Massive and encrusting | Dome: $\frac{1}{6}\pi h(3r^2 + h^2)$ | Dome: $\pi(h^2 + r^2)$ |
| Branching | Ellipse: $\frac{4}{3}\pi\left(\frac{h}{2} \times \frac{x}{2} \times \frac{y}{2}\right)$ | Cylinder with top: $2\pi rh + \pi r^2$ (Multiplied by adjustment factor of 0.44) |
| | *Kiel, Huntington & Miller (2012)* | *Naumann et al. (2009)* |

To quantify the living volume and surface area of massive and encrusting corals, formulas for the volume and surface area of a dome were used, while for branching corals, the volume of an ellipse (*Kiel, Huntington & Miller, 2012*) and the surface area of a cylinder with a top with an adjustment factor from *Naumann et al. (2009)* was used (Table 2). These calculated values were then multiplied by the proportion of coral tissue recorded as "living" to account for partial mortality. This approach is conceptually similar to the methods suggested by *Goergen et al. (2020)* for coral restoration monitoring.

To assess overall coral growth and survival related to treatment, the total living coral surface area and volume were summed on each coral plate for each monitoring timepoint to provide sufficient statistical replicates. For each approximate time-since-deployment period (3, 6, 9, 12, and 19 months), the average living coral surface area and volume per coral plate was compared between treatments using t-tests if the data were normal or non-parametric Wilcox tests for non-normal distributions.

### Fish abundance, biomass, and diversity

Statistical tests to assess change in fish communities were applied following methods in *Aburto-Oropeza et al. (2011)*, which evaluated changes in fish communities after establishment of a marine protected area. Changes in fish biomass, abundance, species richness, and species evenness over time (for each survey conducted at each timepoint and/ or treatment replicate) were assessed at the Arks and Control sites, separately, using ANOVA. For two monitoring timepoints (August 2022, June 2023), at least three surveys were conducted for each treatment (Ark *vs*. Control), therefore providing the minimum sample size required to statistically compare differences in total biomass as well as biomass of each trophic guild between treatments using using t-tests if the data were normal or Wilcox tests for non-normal distributions. Other timepoints had fewer surveys, precluding statistical comparison between treatments.

### Turf and macroalgae growth on coral plates

The initial deployment timepoint was excluded from statistical analysis, as the coral plates were comprised of bare limestone with no growth other than translocated corals. Turf and macroalgae coverage on coral plates at other timepoints were compared using non-parametric Wilcox tests to assess whether the coverage was significantly different based on treatment (Ark *vs*. Control for all plates deployed for the same approximate lengths of time). To test whether ARMS affected the amount of turf and macroalgae cover on coral plates, a t-test and a Wilcox test was used to evaluate turf and macroalgae coverage

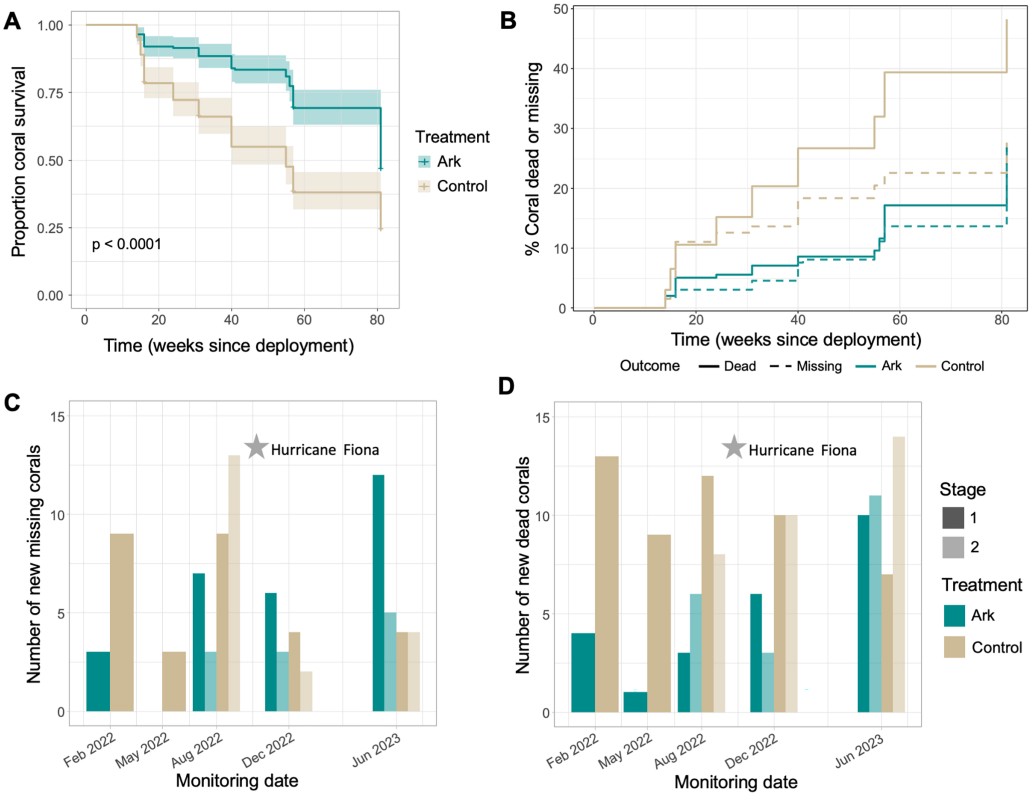

**Figure 2** (Top) Coral survival with time shown as (A) Kaplan-Meier survival curves based on treatment and (B) cumulative risk of either death or falling off coral plates with time based on treatment. (Bottom) Number of new (C) missing and (D) dead corals observed at each monitoring period, colored by treatment and shaded by deployment stage (stage 1 corals deployed November 2021, stage 2 deployed May 2022).

after 3 and 6 months of deployment, respectively, between coral plates that were deployed with (stage 2) or without ARMS (stage 1).

## RESULTS

### Coral survival and growth

After about 19 months, average survival on the Arks was about 47% compared to 24% at the Control sites, with approximately 48% of corals at the Control sites dead and 28% of corals having fallen off plates; in contrast, 28% of corals had died and 26% had fallen off plates on the Arks (Figs. 2A and 2B). Corals were significantly more likely to survive to a given timepoint on the Arks relative to the Control sites (Fig. 2A; Chi-squared = 40.3, $p$ = 2e-10). When death *vs.* falling off was considered, corals were significantly more likely to die at a Control site compared to an Ark after a given amount of time (Figs. 2A and 2B; Gray's test = 23.4, $p$ < 0.001), but there was no difference in the likelihood of falling off of coral plates over time between the Arks and Control sites (Figs. 2A and 2B; Gray's test = 2.7, $p$ = 0.10).

For corals deployed at the same time, fewer corals died on the Arks compared to the Control sites at all monitoring timepoints (except in June 2023, where 10 of the stage 1

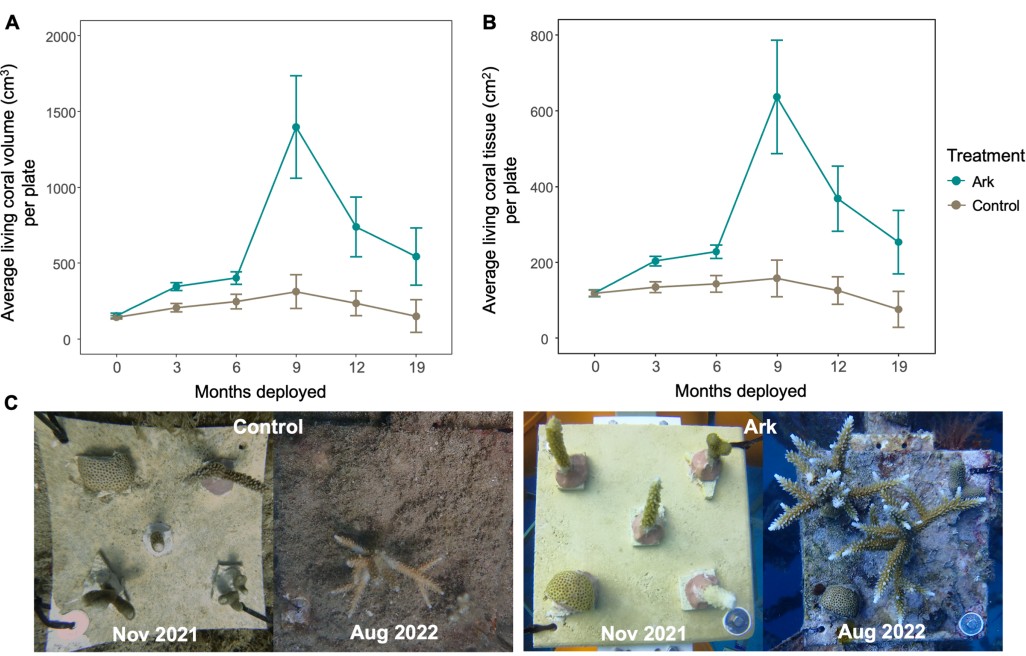

**Figure 3** Average living coral volume (A) and surface area (B) per coral plate, based on the treatment and number of months each plate had been deployed. After month 0, all differences between treatments are significant. (C) Representative photos from a coral plate at a Control site (left) and an Ark (right) at the start of the experiment in Nov 2021 and 9 months later in Aug 2022.

corals initially deployed November 2021 died on the Arks and seven died on the Control sites; Fig. 2D). Corals at the Control sites tended to fall off plates early after deployment, while corals tended to fall off of the Arks after longer periods of time (Fig. 2C). There was no obvious impact on loss or death of corals related to the passage of Hurricane Fiona in September 2022 (Figs. 2C and 2D). Considering both coral survival and living growth, the average living volume and surface area of coral on each coral plate was significantly higher on Arks compared to Control sites at all timepoints (Figs. 3A and 3B; $p < 0.01$ for all comparisons). The largest amount of coral growth was observed after addition of ARMS to the Arks (Figs. 3A and 3B).

## Fish abundance, biomass, and diversity

At the initial timepoint, no fish had yet discovered the Arks structures, and at the second monitoring timepoint (Feb 2022), only a few small fish had begun to associate with the Arks (mostly wrasses and juvenile blue tangs). Total fish numbers and biomass both significantly increased over time at the Arks ($p = 0.003$ and $p = 0.02$, Fig. 4A), while at the Control sites, neither fish biomass nor abundance changed significantly with time ($p > 0.18$; Figs. 4B and 4C). Differences in fish biomass and abundance between treatments could only be statistically compared in August 2022 and June 2023; biomass was not significantly different between treatments, but there were significantly higher numbers of fish associated with the Arks compared with the Control site in August 2022 ($p = 2.119e-05$; Fig. 4C).

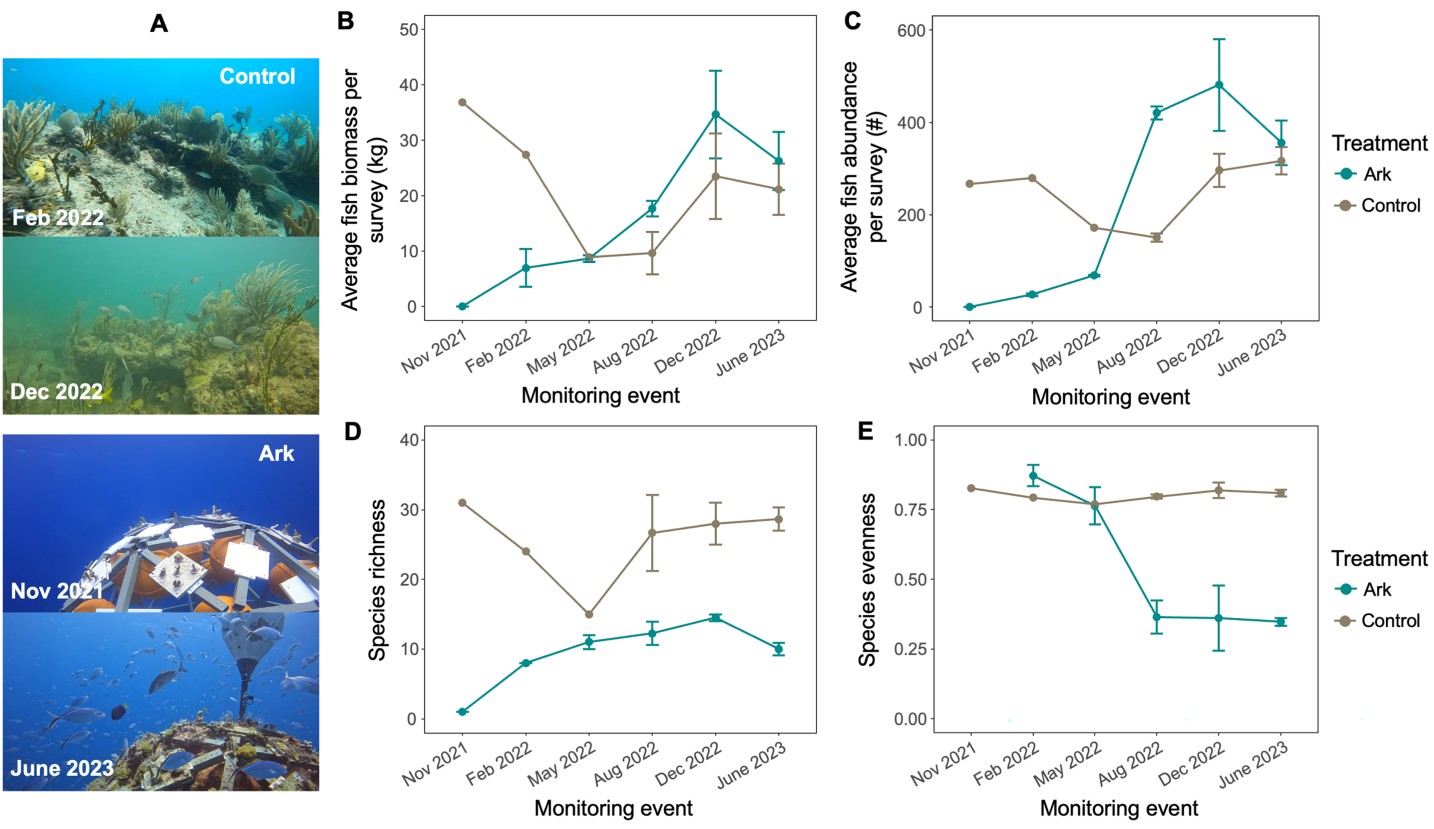

**Figure 4** Fish communities at the Arks and Control sites, with (A) representative photos at two time points. Fish (B) biomass and (C) abundance associated with each treatment at each monitoring timepoint. Fish (D) species richness and (E) evenness associated with each treatment at each monitoring timepoint.

Fish communities associated with the Arks had lower species richness than the Control sites at all timepoints, but species richness increased over time at the Arks ($p = 0.04$), with no significant temporal change at the Control sites ($p = 0.5$; Fig. 4D). Fish species evenness did not change significantly with time at the Control sites ($p = 0.6$), and decreased over time at the Arks (excluding timepoint 0, $p = 0.003$; Fig. 4E), as the fish community became heavily dominated by piscivores (Fig. 5).

The trophic roles of fish associated with the Arks and Control sites changed through time and differed between treatments (Figs. 5A and 5B). In August 2022, 9 months after the Arks were deployed, there was significantly higher biomass and numbers of piscivores at the Arks ($p < 0.04$ for both), and higher numbers and biomass of planktivores at the Control sites ($p = 0.001$ and $p = 0.01$, respectively), with no significant differences in other trophic guilds (Figs. 5A and 5B). In June 2023, about 19 months after the Arks were deployed, there was significantly higher biomass (mean 24.6 kg at the Arks, 4.5 kg at the Controls; Wilcox test $p = 0.028$) and numbers of piscivores (mean approximately 306 at the Arks, 3 at the Control sites) and significantly less biomass and fewer planktivores at the Arks compared to the Control sites (Wilcox tests $p = 0.04$ for both). In addition, there were significantly fewer primary and secondary consumers at the Arks compared to the Control sites (t-tests $p < 0.001$ and $p = 0.036$, respectively), and lower biomass of secondary

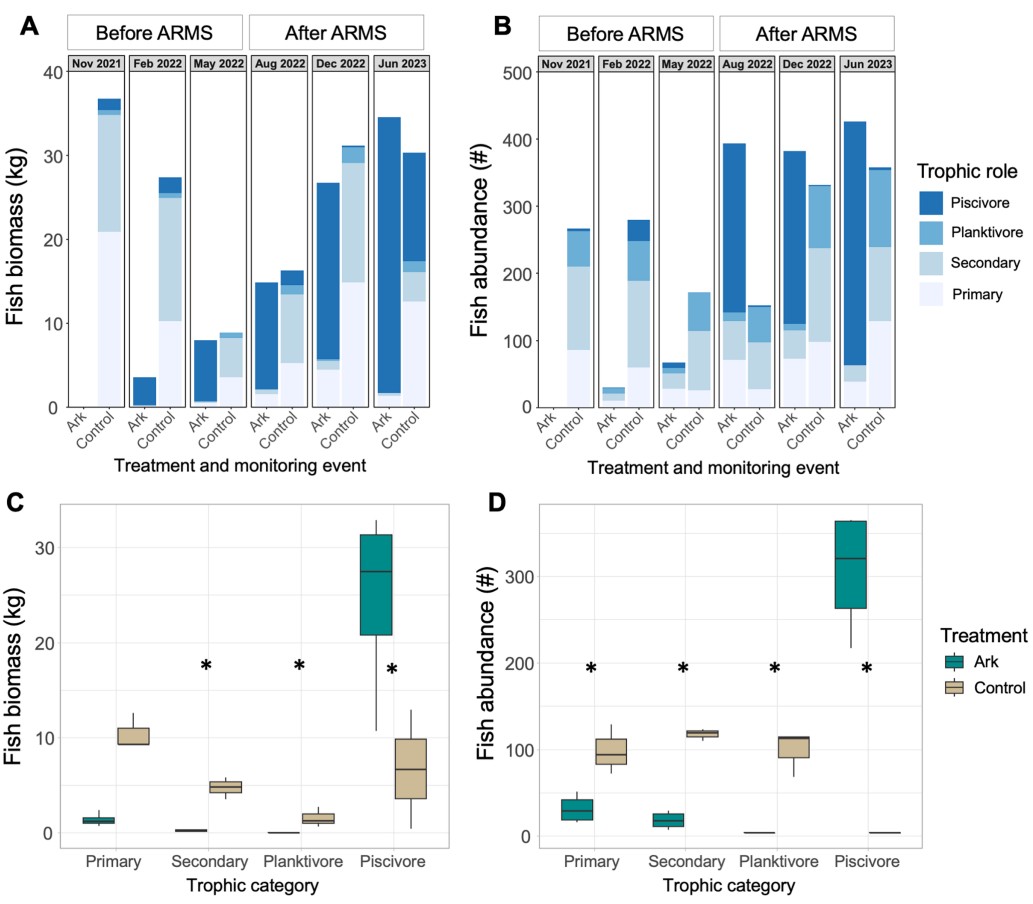

**Figure 5** Fish (A) biomass and (B) abundance from a representative survey from each treatment for each monitoring timepoint, separated by trophic role. (C) Biomass and (D) number of fish recorded in each trophic category within each treatment, from June 2023 monitoring data. Significant differences between treatments indicated with an asterisk.

consumers at the Arks compared to the Control sites (t-test $p = 0.02$; Figs. 5C and 5D). As shown by these results, as well as reduced species diversity and evenness values, the fish community at the Arks is heavily skewed towards piscivorous fishes, with high abundances of bar jacks (*Carangoides ruber*) and almaco jacks (*Seriola rivoliana*) observed associating with the Arks. The number and biomass of piscivores associating with the Arks was significantly enhanced after ARMS were added in May 2022, compared to before the addition of ARMS (t-tests $p = 0.005$ and $p = 0.0002$, respectively; Fig. 5). In contrast, there were no significant differences in biomass or numbers of piscivores associated with the Control sites between these time periods.

## Turf and macroalgae on coral plates

Combined turf and macroalgae cover was significantly higher on coral plates at the Control site compared to the Arks at all timepoints after time 0 ($p < 4.723e-08$ for all comparisons; Fig. 6A). After initial increases 3–6 months after deployment, turf and macroalgae cover significantly decreased over time on the Arks (F-statistic: 6.392 on 1 and

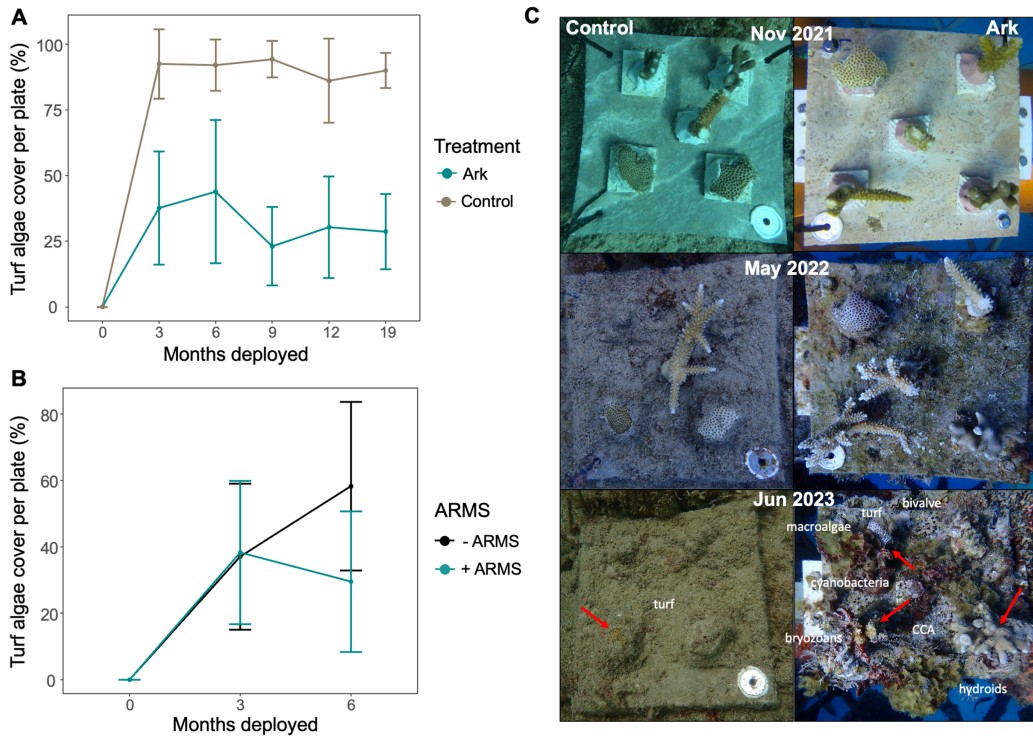

**Figure 6** (A) Average turf and macroalgae coverage on coral plates at each monitoring timepoint, separated by treatment. After month 0, all differences between treatments are significant. (B) Average turf and macroalgae cover per coral plate after the initial 3 and 6 months of deployment for coral plates deployed in the first project stage without ARMS (−ARMS) and in the second project stage with ARMS (+ARMS). (C) Side-by-side comparison of the biofouling coommunities developed over 19 months on a representative (left) Control site coral plate and (right) Ark coral plate. In the bottom (June 2023) panels, remaining living corals are indicated with red arrows. Labels are also included, indicating some of the non-coral organisms visible on the coral plates.

158 DF, *p*-value: 0.01245), with no significant change over time on Control site plates (F-statistic: 2.557 on 1 and 152 DF, *p*-value: 0.1119; Fig. 6A).

Some turf and macroalgae cover were likely removed from Control site plates by sand scouring, while at the Arks, it may have been grazed down and/or overgrown or outcompeted by other organisms such as sponges, fire coral, crustose coralline algae, and bryozoans. These other competing organisms were also observed to overgrow some living corals on coral plates on the Arks (Fig. 6C).

Turf and macroalgae coverage on the Arks coral plates was not significantly different for those plates deployed with or without ARMS after about 3 months of deployment (means of 37% and 38% cover, respectively), but was significantly higher for coral plates deployed without ARMS (mean of 58% cover) than with ARMS (mean of 30% cover) after about 6 months of deployment (Wilcox test, *p* = 0.001; Fig. 6B). These results may be influenced by seasonal changes, as the 6-month timepoint for coral plates deployed without ARMS was May 2022 and with ARMS was December 2022. However, at the Control sites, coral plates deployed at the same times as on the Arks displayed the opposite pattern, with slightly but significantly lower turf and macroalgae cover 6 months after deployment for those plates

deployed in stage 1 (mean of 89% cover in May 2022) *vs.* stage 2 (mean of 95% cover in December 2022; Wilcox test $p$ = 0.02), suggesting the differences in turf and macroalgae cover on coral plates after 6 months on the Arks was associated with the addition of ARMS (Fig. 6B).

## DISCUSSION

Stony corals had better survival and growth on the midwater Arks systems relative to the seafloor at the same depth, demonstrating that environmental conditions on Arks were better for stony corals than conditions on the benthos near Vieques. More broadly, the Arks system outperformed benthic translocation approaches typically used in coral mitigation and coral outplanting, analogous to the improved performance of corals grown in nurseries on structures off the benthos (*Shafir, Van Rijn & Rinkevich, 2006*). Yet, unlike coral nurseries and compared to the Control sites, the Arks also had more predatory fish, lower levels of turf and macroalgae overgrowth, and qualitatively higher biodiversity. Higher levels of coral survival may be related to favorable environmental conditions such as higher dissolved oxygen, fewer bacteria and more viruses, higher water flow speeds, higher light intensity (*Baer et al., 2023*), and/or improved ecological function at the Arks sites. These characteristics indicate that the Arks developed a self-sustained reef ecosystem, favor coral over macroalgae, and generate enhanced ecosystem services compared to the natural reefs from which they were seeded.

A meta-analysis of coral restoration projects worldwide found an average survival rate of 66% for translocated corals, though this rate does not take into account differing lengths of time that various projects were monitored (*Boström-Einarsson et al., 2020*). At the Control sites, 66% of corals survived for about 8 months (31 weeks; Fig. 2), but after that time, survival continued to decline, with just 24% of corals still alive after about 19 months (Fig. 2). On the Arks, about 69% of corals were still alive after more than a year (57 weeks), indicating about 50% longer survival compared to the Control sites, and 47% of corals were still alive after about 19 months (about twice as many remaining live corals as at the Control sites; Fig. 2). These data show that assessments of coral transplantation projects should establish a "local background" survival rate for translocated corals, as in the Control sites used here, to fully assess the efficacy of a given approach. Survival of corals on the Arks was lower than survival in a 2007–2009 study in Vieques which also translocated corals to artificial reef structures (73% survival to 19 months; *Dial Cordy and Associates Inc, 2013*). That study used larger colonies rather than fragments (*i.e.*, more robust stock) and took place more than 15 years ago, during which time there have been multiple mass coral bleaching events and the emergence of new coral diseases in the Caribbean.

Coral translocation creates the potential for coral loss through detachment (epoxy attachment and entire fragment falls off) or breakage (portion of coral fragment breaks off), as well as coral loss due to mortality. The rate of detachment was not statistically different between the Arks and Controls and was similar to rates of detachment reported elsewhere (*i.e. Dizon, Edwards & Gomez, 2008*). Incidental grazing disturbance by herbivorous fishes can cause detachment of experimental coral nubbins (*Quimpo, Cabaitan & Hoey, 2020*), and this effect may explain the larger loss of corals from Control
site plates within the first 3 months of deployment, given that very few fish were observed at the Arks during this time period. Interestingly, relatively few corals (9) fell off the Arks during the time period that Hurricane Fiona passed almost directly over the Arks (September 2022), suggesting coral loss was not strongly tied to storm events. The rate of corals falling off of the Arks generally increased over time, possibly because as corals grew larger, they became more top-heavy and detached more easily, or their increased size created stronger horizontal drag forces that allowed currents to dislodge the corals (*Madin & Connolly, 2006*). However, breakage of corals off of Arks structures is not necessarily problematic; breakage can facilitate reef substrate accumulation and carbon sequestration on the benthos below an Ark in deep water and/or aid in asexual reproduction of corals from Arks in water shallow enough for coral survival.

The superior performance of corals translocated to the Arks relative to the Control sites was likely the result of direct effects of algal competition and indirect effects of fish communities and microbial processes. Previous benthic artificial reefs built in Vieques found, over a similar period of time, that the reef became covered in turf algae that surrounded the corals (*Dial Cordy and Associates Inc, 2013*). A similar successional trajectory was observed here: Control site coral plates became fouled almost exclusively by turf and macroalgae (as well as sediment bound to these substrates) that surrounded the coral fragments and remained this way throughout the study. In contrast, fouling communities surrounding coral fragments on Arks plates were more diverse, with higher proportions of other invertebrates and lower coverage of turf and macroalgae (Fig. 6). Competition is high on coral reef benthos and turf and macroalgae are some of the strongest competitors of corals, explaining why coral nurseries routinely manually remove algae to support coral growth (*Shafir, Van Rijn & Rinkevich, 2006*). No algal removal was completed on the Arks, though, allowing the system to develop relatively naturally into a complex midwater reef system instead of a maintained nursery. Instead, higher diversity reef communities formed, enhanced by the addition of ARMS, which was associated with decreases in turf algal cover and increases in species diversity with time.

The Arks developed a piscivore-dominated fish community with numbers and biomass of fish associated with the Arks similar to or greater than the Control sites (Fig. 4), particularly for fishery target species such as jacks. Top-heavy, piscivore-dominated coral reef food webs, as observed on the Arks, are typically associated with low standing stock of algae and herbivores, as trophic efficiency is high (*Sandin et al., 2008*). Higher cover of turf and macroalgae are strong predictors of poor reef health and "microbialization" (*Haas et al., 2016*; *Silveira et al., 2023*), likely due to algae releasing dissolved organic matter that bacteria feed upon and draw down dissolved oxygen (*Mueller et al., 2022*). The Control sites had lower dissolved oxygen, more bacteria and fewer viruses, lower water flow speeds, and lower light intensity despite similar depths than the Arks (*Baer et al., 2023*), demonstrating additional indirect effects pushing the Arks system towards corals winning over algae.

## CONCLUSIONS

While small in size, Arks provide numerous ecological benefits and ecosystem services. Arks increased survival and growth of translocated corals, suggesting these systems could be used for mitigation and to enhance restoration projects. Specifically, higher coral survival and the presence of multiple coral recruits on the Arks suggests they could act as a source of larvae to nearby reefs (*Amar & Rinkevich, 2007*). Top-heavy fish communities, particularly after addition of seeded ARMS, highlight that Arks can enhance fisheries productivity. The addition of seeded ARMS was associated with lower turf abundance. While not quantified during limited monitoring events for this project, many juvenile fishery target invertebrates including scallops, lobster, and crabs were also observed on the Arks. Arks can therefore act as *in situ* mesocosms for scientific studies (*Baer et al., 2023*), "house reefs" for divers, snorkelers, and education, and can contribute to coral reef mitigation and restoration.

Arks create the opportunity for ecosystem-scale tests of coral reef restoration strategies and can be used to measure the response of these complex ecosystems to perturbations *in situ*. The replicability of Arks can increase statistical power and inference. The geodesic Ark design could be further developed to suit a variety of questions and needs. For example, the surface area of the structure could be increased by placing panels or baffles on the struts to provide more space for settlement and growth of organisms or to purposefully direct or retain water within the structure. Some studies may benefit from a scaled down or smaller Ark design to ease deployment, enhance replicability, and allow for greater manipulation of the system, including tests involving moving Arks between locations. Systems to manipulate the distance of Arks from the benthos (*e.g.* winches) may be included to, for example, draw corals to lower temperature and light at depth during warming events and protect them from bleaching. Lastly, better understanding of systems-level dynamics could be further enhanced by adding more numerous sensors throughout the structure (*e.g.*, oxygen, flow, *etc.*) for higher-resolution understanding of ecosystem behavior.

## ACKNOWLEDGEMENTS

We are grateful for the support received from Dan Waddill, Kevin Cloe, Daniel Hood, Maria Danois, P.F. Wang, Adam Candy, John Martin, Lora Pride, Brett Doerr, Ronny Fields, Kristin McClendon, Nilda Jimenez Marrero, Michael Nemeth, Tali Vardi, Sarah Elise Field, Pedro Rodriguez, Pete Seufert, Tania Puell, Megumi Kirby, and many others, particularly from Jacobs Engineering and Vieques too numerous to name.

### Funding

This work was supported by the United States Department of Defense Environmental Security Technology Certification Program (No. CR20-5175 to Jessica Carilli), the Gordon and Betty Moore Foundation (No. 9207 to Forest Rohwer), and the National Science Foundation (No. 2022717 to Aaron C. Hartmann and No. 2209377 to Mark Little). The

funders had no role in study design, data collection and analysis, decision to publish, or preparation of the manuscript.

### Grant Disclosures

The following grant information was disclosed by the authors:
United States Department of Defense Environmental Security Technology Certification Program: CR20-5175.
Gordon and Betty Moore Foundation: 9207.
National Science Foundation: 2022717 and 2209377.

### Competing Interests

Jessica Carilli and Gunther Rosen are civilian employees of the US Department of the Navy. Bart Chadwick is Principal and President of Coastal Monitoring Associates.

### Author Contributions

- Jessica Carilli conceived and designed the experiments, performed the experiments, analyzed the data, prepared figures and/or tables, authored or reviewed drafts of the article, and approved the final draft.
- Jason Baer conceived and designed the experiments, performed the experiments, analyzed the data, prepared figures and/or tables, authored or reviewed drafts of the article, and approved the final draft.
- Jenna Marie Aquino performed the experiments, authored or reviewed drafts of the article, and approved the final draft.
- Mark Little performed the experiments, analyzed the data, authored or reviewed drafts of the article, and approved the final draft.
- Bart Chadwick conceived and designed the experiments, performed the experiments, authored or reviewed drafts of the article, and approved the final draft.
- Forest Rohwer conceived and designed the experiments, performed the experiments, authored or reviewed drafts of the article, and approved the final draft.
- Gunther Rosen conceived and designed the experiments, authored or reviewed drafts of the article, helped secure permissions to conduct project, and approved the final draft.
- Anneke van der Geer performed the experiments, analyzed the data, authored or reviewed drafts of the article, and approved the final draft.
- Andrés Sánchez-Quinto performed the experiments, authored or reviewed drafts of the article, and approved the final draft.
- Ashton Ballard performed the experiments, authored or reviewed drafts of the article, and approved the final draft.
- Aaron C. Hartmann conceived and designed the experiments, performed the experiments, analyzed the data, prepared figures and/or tables, authored or reviewed drafts of the article, and approved the final draft.

### Field Study Permissions

The following information was supplied relating to field study approvals (*i.e.*, approving body and any reference numbers):
Coral collections were conducted under Puerto Rico Department of Natural and Environmental Resources permit number O-VS-PVS15-SJ-01233-20092021. Field experiments were conducted with approvals under the National Marine Fisheries Service Endangered Species Act Section 7 Programmatic Biological Opinion on the Underwater Investigation and Removal/Remedial Activities in UXO 16, Vieques, Puerto Rico, OPR-2017-00026 and associated consultations.

## Data Availability

The raw data and the R code are available in the Supplemental Files.

## Supplemental Information

Supplemental information for this article can be found online at http://dx.doi.org/10.7717/peerj.17640#supplemental-information.

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
