# Peer review of "Escaping the benthos with Coral Reef Arks: effects on coral translocation and fish biomass"

_PeerJ, doi:10.7717/peerj.17640_

## Round 0.1 · original submission · Major Revisions

Apologies for the late feedback. It was quite difficult to find reviewers willing to review the manuscript, and those who agreed took longer than expected. On the positive side, the comments given are thorough and significant. One of the comments from the reviewer that I think you need to think about is the microbialization concept, which he/she believes is not suitable for this manuscript.

I suggest that you adhere to the suggestions and send me the revised version as soon as possible.

Reviewer 1 ·

Basic reporting

no comment

Experimental design

no comment

Validity of the findings

no comment

Additional comments

Overall comment:

The article presents a concise overview of the issue of microbialization on coral reefs and explores the hypothesis that relocating corals to midwater geodesic spheres, known as Coral Reef Arks, could enhance the effectiveness of coral restoration attempts. Overall, the research is sound and addresses an urgent matter in the rehabilitation of coral reefs.

Given the article's well-crafted nature, I will offer overarching feedback to enhance the article's overall quality.

Although the paper focuses on the advantages of Arks in avoiding the effects of microbialization, it does not provide specific information about the environmental characteristics related to the microbialization effects that differ between Arks and the Control group. Providing further information on these elements is crucial to showcasing the study's strength and reliability. Thus, it is advisable to include further information regarding the environmental parameters to address this concern.

Including information regarding potential constraints and future avenues for investigation would be advantageous in order to provide a comprehensive outlook on the study's implications.
Although the conclusion emphasizes the present advantages of Arks, it would be advantageous to include a concise remark to potential future prospects or suggestions for enhancing Ark design, deployment techniques, or more research opportunities in order to fortify the conclusion. This has the potential to serve as an impulse for promoting additional investigation in the field.

(line 400–401) does not seem to be connected with the narrative of the findings and conclusion; thus, I would suggest the author remove the sentence “further enhancing fisheries."

Reviewer 2 ·

Basic reporting

Overall the manuscript is written very well;

Experimental design

There are questions posed about experimental design (see below) that need better justification

Validity of the findings

Findings are interesting, though comparison of metrics between sites needs clarity as the design is uneven. It appears there are different coral assemblages between Arks and control sites? Provide details as this can have the largest influence on survival/health metrics measured.

The control sites are really not valid reference for the Arks due to location and environmental parameters being vastly different.

Additional comments

Overall;
This study investigates reef assemblages translocated on to midwater geodesic spheres (called Coral Reef Arks) compared to control sites fixed to a patch reef benthos. The metrics of coral survival, living tissues surface area, fish abundance and biomass plus algae and turfs algae cover between Arks and control sites were compared. Overall, the study is relatively well written and so have few issues or points with the overall results and presentation of the data. However, there are a couple of critical issues with the experimental design that should be justified or clarified.
One overall comment would be that links are made to microbialization as the driver for better performance of the Arks over the control deployments. However, the narrative is difficult to support as there is no information provided on microbialization at the sites and as the driver for differences in metrics measured between Arks and controls installations. I would suggest removing these links to microbialization unless these metrics are measured/presented. The finding of improved survival are somewhat to be expected and this is actually highlighted in the introduction; see line 52 onwards where coral nurseries have repeatedly reported better coral survival metrics when elevated off the benthos as they avoid competitive interactions that result in high mortality. The novelty of this paper is just a different design for restoration activities, that has some benefits likely through a range of processes including reduced competition better water flow (and maybe microbialization effects, though that is speculative).
There needs to be justification for why the control sites are so far removed from the location of the Arks. The control sites really are not similar in any way to the environment surrounding the Arks and while this is a point being made by the manuscript some discussion and qualification is required. The only justification provided is that the control sites are at the same depth, though control sites are much closer to land-based areas and potential impacts with the Arks being in more oceanic condition by what is presented in the manuscript. Hence a whole raft of environmental conditions would be different at the two sites and confounds that comparisons potentially. Why not have the control installation at the same areas where the Arks are deployed (depth being the only major change perhaps between treatment and control). Overall there needs to be more context in the discussion detailing what environmental conditions (sedimentation, currents, light etc) that may influence the results overall.
There is no detail of what coral species are deployed onto the Arks and what species are deployed at the control sites (unless I missed it somewhere). Figure 1 indicates 4 coral spp. on the Arks while 8 spp. on the control sites. This seems an uneven design and so metrics of coral mortality, live coral tissue etc are highly dependent on species and number of species deployed on the devices. Table 2 does provide the way live surface area was calculated for massive, encrusting and branching species, though data outcomes metrics are not broken down to that level. Branching corals maybe more likely to be dislodged than massive species; though there is no information of what number of each species/morphologies are deployed across the various sites. All survival/health metrics would seem to be highly dependent on species composition and so for relevant comparison of Ark performance and controls sites these comparisons need to be made. As an example, the dislodged corals maybe more likely branching species, due to their morphology, though there is no information if there are branching species at one site and not the other, or more individuals at one site vs the other. Rather than generalized mortality, and growth data averaged over all species/morphologies; these metrics need to be broken down and displayed individually I would think to give comparative assessments.
Often restoration manuscripts request the costing of the approaches for comparative assessments between methods. This can be difficult and not advocating it is needed in this case though some mention of costs/logistics for these Arks vs other approaches in perhaps needed in the discussion (though this may have been presented elsewhere). The cost of these devices, labor intensive activities of attaching fragments to the Arks and the ability to deploy enough of them to provide ecological services can be debated in line with other restoration activities that use similar approaches.

Other Points
• Title; remove the context of microbialization from the title; as this is not investigated or presented in the manuscript at all.
• Abstract – again remove first 2 sentences as microbialization, microbial loads, oxygen, disease etc are not the focus of the paper and so should not be the scene setting introduction. Both the abstract and title give a misrepresentation of what the paper is about. Aspects of microbialization can be discussed (in the discussion) but removed as the focus of the paper.
• Abstract last sentence; cannot be justified that 2 deployed Arks enhanced the fishery? They likely have a role as a fish aggregation device and enhance biomass but assume there is no fishery per se at these sites.
• Line 129-131: I did not think Acropora cervicornis was susceptible to SCTLD. Perhaps this is not being inferred and it was other species though this is the only listing of corals used in the experiments that I can see.
• Not a fish expert – but what is the background fish populations without these installations. I agree the first time points may provide that background and the Arks had few fish but that is likely since midwater there are few fish unless there are aggregating devices which these Arks provide. Are the fish drawn away from other habitats nearby though rather than increasing the biomass of the area “fishery”.

---

## Round 0.2 · Minor Revisions

Your responses to the reviewers' comments are sufficient. I understand one of them's concerns about the selection of the control site, but in reality, it is quite difficult to find an ideal site with parameters similar to those of the experimental location.

I have sent your revised manuscript for a second review. While the reviewer is satisfied with the overall response and improvement on the manuscript, the explanation/justification on the control (or the experimental site) is still not in the reviewers' favour. I think we can agree to disagree with the reviewer's opinion. Nevertheless, if you could respond to the reviewer's comment, ' Having Arks mid-water in nearshore reef areas at the control sites would control for that and could show similar coral metrics due to escaping competitive processes on the benthos.' it would serve justice to your study.

In addition, there are minor issues that I hope you can address in the manuscript.
- throughout the manuscript, you use the term 'benthos' as a non-living entity. For example, 'Coral plates were secured to either one of the Arks or to the benthos at one of the Control sites using stainless steel hardware and/or cable ties (Fig. 1C).' (lines 162-163). My understanding (at least myself) is that benthos refers to organisms that live on (epifauna) or in (infauna) the sediment. Does 'benthos' in your manuscript refer to coral (organisms) or hard substrate (non-living)?
- Based on Baer et al. (2023) and this manuscript, the use of Coral Reef Arks varies, ranging from mesocosms study to enhancing conservation projects. In addition, the Ark provides a permanent structure for transplanted corals as per the statement: 'In contrast to growing corals in isolation for short periods, Arks are intended to provide the same or more beneficial water quality conditions as nurseries while creating an artificial reef for corals to permanently reside.' (lines 57-59). However, looking at the structure of the ark (spherical), perhaps you can explain in the Introduction how the Ark will be placed on the degraded reef as part of the conservation effort, or is the ark intended to remain buoyant indefinitely?
- The control site does not have ARMS units in close proximity as Arks. Perhaps you could explain the reason for this to avoid confusion.
- Typographical errors in the supplementary file (list of coral species), the '.sp' should not be italicised.

Looking forward to your final version.

Reviewer 2 ·

Basic reporting

Overall the manuscript is well prepared

Experimental design

Good design and clearly laid out

Validity of the findings

Finding valid, interesting and well presented

Additional comments

The revised manuscript of Carilli et al. has addressed well all the comments raised in the review process. As highlighted previously the work investigates reef assemblages translocated on to midwater geodesic spheres (called Coral Reef Arks) compared to control sites fixed to a patch reef benthos. The manuscript is well constructed, well written, data robustly analyzed with nice images supporting the story. Overall, the authors have addressed all the major points raised and it is acceptable for publication in my opinion. It represents a nice addition to the field and explores alternative approaches to in the space on boosting resilience for coral reef ecosystem.

My only comment would be I am not totally sold on the justification for controls sites verses deployment sites of the Arks. I understand that Arks deployed in mid-water areas with better water quality can have great results for coral survival and development of reef communities. This current study is however directly comparing coral survival and other macroorganisms metrics between control sites and Arks. To be totally comparable it seems to me you would need to deploy Arks and control sites in the same areas with similar water quality and environmental parameters. Having Arks mid-water in nearshore reef areas at the control sites would control for that and could show similar coral metrics due to escaping competitive processes on the benthos. Perhaps one sentence in the discussion can handle that question.

---

## Round 0.3 · accepted · Accept

All comments from the reviewer have been addressed satisfactorily. I think additional sentence (lines 299-300 in the track changes pdf) is needed to emphasize the benefits of the ark. So let's keep it permanently.

There is however a very tiny typo (line 299) which I believe is '=' but accidentally replaced with 'ì'. I think you can change that in the galley proof session. It will hold your manuscript much longer if I return back to you to correct it at this stage.

Other than that your manuscript is ready for publication. Hooray!